# “I Genuinely Believe This Is the Most Stigmatised Group within the Social Care Sector”—Health and Social Care Professionals’ Experiences of Working with People with Alcohol-Related Brain Damage: A Qualitative Interview Study

**DOI:** 10.3390/ijerph21010010

**Published:** 2023-12-20

**Authors:** Peter Johan Kruithof, William McGovern, Catherine Haighton

**Affiliations:** Department of Social Work, Education and Community Wellbeing, Faculty of Health and Life Sciences, Northumbria University, Newcastle upon Tyne NE7 7XA, UK; peter.j.kruithof@northumbria.ac.uk (P.J.K.); william.mcgovern@northumbria.ac.uk (W.M.)

**Keywords:** stigma, alcohol-related brain damage, qualitative research, health and social care professionals

## Abstract

Appropriate diagnosis, treatment and care contribute to better service engagement, improvements to wellbeing, cost savings and reductions in morbidity and mortality for people with alcohol-related brain damage. In Northeast England, large amounts of alcohol are consumed; this is reflected in the number of alcohol-related deaths in the region. However, the pathway for people with alcohol-related brain damage to receive diagnosis, treatment and care is unknown and could be unwittingly influenced by stigma. Qualitative, in-depth, semi-structured interviews were completed with 25 health and social care professionals from organizations involved with people with alcohol-related brain damage recruited via snowball sampling. Interviews were recorded, transcribed verbatim, coded, and analysed. People with alcohol-related brain damage were found to be stigmatised by both society and professionals, inhibiting their entry into services. Therefore, alcohol-related brain damage remains underdiagnosed and misdiagnosed. There was found to be no dedicated service; silos with revolving doors and underfunded generic care with long waiting lists typically exclude those with alcohol-related or neurological problems. Reducing stigmatising processes associated with alcohol-related brain damage could counteract professionals’ reluctance to provide care.

## 1. Introduction

In Northeast England, large amounts of alcohol are consumed, with adults in the region most likely to drink more than 14 units of alcohol per week (29%) compared to all other regions in England [1]; this is reflected in the large number of alcohol-related deaths [1]. In 2021, there were 7556 alcohol-specific deaths in England, equating to a rate of 13.9 per 100,000 population, with a mortality rate that is the highest in the northeastern region (20.4 per 100,000 population) [2]. Excessive alcohol consumption, in combination with poor diet, can result in alcohol-related brain damage (ARBD)—an umbrella term referring to the neurocognitive impairments caused [3]. ARBD commonly describes Wernicke encephalopathy, Korsakov syndrome, alcohol-related dementia and other alcohol-induced disorders that negatively influence the structure of the brain [4]. The symptoms of ARBD vary but include problems with cognitive functioning and memory alongside physical symptoms. Without diagnosis and care, people with ARBD suffer substantial morbidity and mortality [5], resulting in significant personal, economic and societal costs [6]. ARBD is also compounded by health inequality, with the most disadvantaged in society suffering the most severe consequences [7]. The number of people in Northeast England with ARBD is hard to determine. However, Northeast England extends to the Scottish borders and the literature from Scotland has reported the prevalence of ARBD among homeless hostel dwellers to be as high as 21% [8], while data from South Wales in the UK report a more conservative prevalence of 0.03% in the general community [9]. The literature suggests that this may be due to ARBD being a difficult disease to diagnose, as not all the clinical signs are always present [10]; meanwhile, other research suggests that health professionals do not always enquire about their patient’s alcohol use [11]. In the UK, there are no national guidelines or standardised health and social care pathways for people with ARBD [12].

Increasingly, stigma is conceptualised as a form of power used to discipline and exploit people [13]. It can result from social, public, or private processes and can operate at micro and macro levels and in a manner that serves the interests of dominant and privileged individuals and organizations [14]. Stigma is theorised in many ways in health and social care settings, but it essentially equates to a process whereby individuals are devalued because they are perceived to be unable or unwilling to adhere to their societal commitments and role obligations. In particular, it is often those subgroups of corresponding individuals like problematic and dependent alcohol users who find themselves subject to stigma and negative judgements being made about their character, motivations and individual abilities [15]. More specifically, stigma has been reported to harm mental health, inhibit access to services, increase feelings of isolation and corrode a person’s sense of self-worth as a valued human being [16]. For example, stigma and negative attitudes towards people with alcohol use disorders in general have been highlighted in the previous literature as a significant barrier to treatment [7]. The literature suggests that patients with alcohol use disorders face stigma and condemnation, with some clinicians holding the view that alcohol-related illnesses are self-inflicted and hence are not worthy of medical care [7]. Therefore, stigma related to alcohol use disorders in general also contributes to making individuals with ARBD a hidden population [17]. However, reviews of the literature have postulated that people with ARBD are subject to the dual or double stigma of an alcohol use disorder in addition to a cognitive impairment [18,19].

A review of the international literature of the needs of people with ARBD reported that, when people with ARBD were asked how they were perceived by the wider community, they considered that others distinguished them as “sub-human” and reported negative feelings [18]. This perception often arose from previous negative experiences of being isolated and ostracised because of what they described as past poor behaviour during years of excessive alcohol use. The review concluded that these experiences reflected the inflexible and regimented healthcare services which only provided rudimentary treatments and ignored individual needs [18]. Indeed, those with ARBD are often viewed as non-compliant or problematic patients due to their cognitive impairments [20].

A rapid review, aimed at exploring the experience of living with ARBD and the associated treatment or support offered, reported services for those with cognitive impairment under the age of 65 years often excluded those whose impairment was related to alcohol and that the experience of living with alcohol-related brain damage and engaging with services is poorly understood [19]. In a recent mixed-method evaluation of an online ARBD awareness-raising programme for frontline care and support practitioners, a number of participants reported the potential overlap of ARBD and other conditions, particularly forms of dementia [21]. These participants expressed their concerns that this may mean that these service users were not receiving the correct support and treatment for their condition, which may be exacerbated by people feeling ashamed of disclosing information about their drinking habits and the associated stigma [21]. This work also alluded to the stigma which was often associated with ARBD and alcohol use disorders. Participants reported that there were often challenges in making referrals and receiving a diagnosis for ARBD, although stigma was specifically perceived as a barrier to service users receiving appropriate treatment. Some participants also raised concerns that service users may not receive the support that they need due to stigma from medical professionals [21].

There is, however, a paucity of literature addressing either experiences of people living with ARBD, or of those providing services and support to that population [19]. In their review, Schölin et al. [19] concluded that understanding these life experiences should be a research priority in order that future service provision, support and interaction can effectively meet the needs of individuals and their families. McCabe interviewed 12 care home staff from ARBD specific care homes in Scotland and reported that staff had a poor understanding of the condition and had received no formal training on ARBD [22]. More recently Heirene et al. interviewed 17 Welsh health and social care professionals supporting people with alcohol-related neurocognitive disorders (ARND), who reported the challenging behaviours associated with people with ARND, their significant physical and mental health needs, stigmatization from others and a general lack of understanding and awareness of ARND [23]; meanwhile, van den Hooff and Goossensen were able to explore the experiences of six Dutch patients suffering from Korsakoff’s syndrome, concluding that the received care was not always perceived as good [24]. However, the aim of this current study was to investigate health and social care professional’s experiences of working with people with ARBD in Northeast England. Reducing the stigma and stigmatising processes associated with alcohol-related brain damage could improve care for people with alcohol-related brain damage.

## 2. Materials and Methods

### 2.1. Design

A qualitative semi-structured in-depth interview study with health and social care professionals from organizations involved with people with ARBD using a Straussian grounded theory approach [25]. Reported according to the consolidated criteria for reporting qualitative research checklist [26].

### 2.2. Ethics

This study received Northumbria University Ethical approval (ref. 9271), Health Research Authority and Health and Care Research Wales approval (REC ref. 19/HRA/0711; Protocol Number 9271), Caldecott approval (01/11/19) and organizational approval (where applicable).

### 2.3. Participants, Setting and Sampling

Eligible participants were any health and social care professionals from organizations involved with people with ARBD, including the National Health Service (NHS), the social care sector, the Voluntary and Community Sector (VCS) and Public Private Partnerships (PPP), in Northeast England, the region of the UK with the most alcohol-related deaths [27]. Snowball sampling, where current participants identify others who might be appropriate and eligible future participants, was used. However, this was supplemented with purposive sampling to target professionals from organizations involved with people with ARBD who were not identified via snowball sampling. The sample size was defined a priori using the model of information power [28], as opposed to empirically through data saturation. The sample size was estimated from the aim, the specificity of the sample, the use of theory, the interviews and an analysis strategy. As the aim was relatively specific, snowball sampling was supplemented with purposive sampling; the analysis used an established theoretical framework; the authors were experienced in qualitative research and aimed to build a rapport with participants before interviewing; the estimated total required sample size was around 20 participants.

### 2.4. Recruitment and Consent

Academics and local investigators, with key contacts in the alcohol field, identified the first eligible participants. Eligible professionals were introduced to one of the authors (P.K.) via email where they would be given a brief explanation of the research, provided with a participant information sheet (PIS), given the opportunity to ask questions and invited to participate in an interview. In most cases the professional was willing to be interviewed and so an appointment was made for (originally) a face-to-face interview or (following COVID-19 restrictions) a telephone/online interview. Following each interview, the participant was asked whether there were any other professionals or organizations that they would recommend for an interview. This approach facilitated access to connected professionals who, while holding different roles, experiences and views, worked with the same types of clients with the same sorts of needs. Participants provided informed written consent prior to each interview and were provided with a debrief sheet after each interview with details of how to withdraw their interview data should they decide to.

### 2.5. Interviews

Interviews were conducted between 17 December 2018 and 12 February 2022 by one of the authors (P.K.), a male researcher who had previous experience of interviewing health and social care professionals in the alcohol field. The author introduced himself as a PhD student and had no prior relationship with the interviewees. The interviews were based on a flexible topic guide which was developed via the literature and through author discussion (see Appendix A). Questions focused on participants understanding of ARBD, process of diagnosis (if appropriate) and care provided for people with ARBD. During the interview process, demographic data were collected, indicating the participant’s gender, age, organization type, role, number of years’ experience in the field and qualification. A pilot interview, with a colleague with a background in social work, provided improvements to the sequencing and wording of probes. Face-to-face interviews were held at a time and place convenient to the interviewee either on the university campus, or at the interviewee’s place of work. All interviews were digitally audio recorded and then transcribed verbatim. Transcriptions were fully anonymised and audio recordings deleted upon transcription. Participants were identified by using an ID number. Interviews lasted between 15 and 90 min. While the majority of interviews were carried out on a one-on-one basis, several individuals requested to be interviewed either in pairs or as a group, in order to minimise the impact on day-to-day activities. Field notes were also used to help explain the transcriptions and support analysis.

## 3. Analysis

A Straussian grounded theory approach to analysis was taken [25]. In Straussian grounded theory, the coding process begins with open coding; the analytic process through which concepts are identified and their properties and dimensions are discovered in the data, via line-by-line coding. Each transcription was checked against the audio recording, prior to its deletion, to ensure accuracy. Transcripts were read and re-read a number of times to ensure familiarity. Line-by-line coding was carried out on each transcript by one of the authors (P.K.) as an iterative process to allow analysis to inform the subsequent interviews. Once the transcripts had been subject to open coding, this was followed by the axial coding phase, focusing on combining categories and subcategories to reassemble the data that had been fractured by open coding and selective coding, the process of integrating and refining the theory. P.K. took the lead with axial coding but was supported with regular input, discussion, and agreement from the rest of the study team to agree the codes. Finally, a conditional matrix was developed to visually organise the categories and subcategories of data, and to identify relationships and patterns between them. This conditional matrix was reviewed at regular team meetings and was refined as interviews progressed and further evidence was gathered (see Table 1 for an excerpt from the conditional matrix). All data analysis was conducted via NVivo.

## 4. Results

Twenty-five participants (9 men and 16 women, age range 22–62 years), who worked or volunteered in private, public and charity sector organizations across Northeast England, were interviewed. Years of experience ranged from 1 to 28 years while the participant’s educational background varied. Participant organization type and role are detailed in Table 2 to provide context while also maintaining anonymity. Four relevant themes are presented along with illustrative quotations.

Theme 1: People with ARBD are stigmatised, which inhibits entry into the health and social care system.

Most health and social care professionals reported how people with ARBD are severely stigmatised. This stigma is exhibited by society in general:
*“I genuinely believe this is the most stigmatised group within the social care sector…I had no idea what stigma and discrimination were until I came to work purely drug and alcohol. It doesn’t have any social sympathy attached to it.”*(Participant #3; Social Worker (Substance Use); Council)

But also, they are stigmatised by the professionals that were supposed to care for them:
*“And I think that’s all bound up with this, the judgement around people’s drinking habits and so these GP’s then make their own moral judgements. It is flabbergasting because it can prevent lots of damage. At that point I come in and…just be quite kind of firm with the GP and just making the GP maybe think about “why have you made this decision” you know?”*(Participant #22; Clinical Psychologist (Neuropsychology); NHS)

Stigma was identified by many health and social care professionals as a barrier for people with ARBD to receipt of appropriate diagnosis, treatment and care. As a result, despite there being many entry points into the health and social care system for people with ARBD, some would be denied treatment:
*“I worked with a client that subsequently didn’t turn out he had Korsakoff’s, but for 10 years he suffered extreme memory issues backwards and forwards to general practitioner was told when you’ve stopped drinking, your memory issues will subside…I ended up placing him in a residential detoxification and rehabilitation unit where after three months there was no significant improvement with his memory. I asked for some medical investigations. It turned out he had a brain tumour.”*(Participant #3; Social Worker (Substance Use); Council)

Some health and social care professionals reported that stigma caused people with ARBD to be less inclined to admit to having a problem and therefore less inclined to seek help. In addition, some health and social care professionals described how people with ARBD could become isolated and abandoned:
*“Because the other thing is the isolation of a lot of these people. So there are families where they are very tolerant and stay involved. But…on the whole they tend to be deserted by everybody.”*(Participant #23; Clinical Psychologist; Acquired Brain Injury Rehabilitation Support Service)

Those health and social care professionals who regularly worked with people with ARBD called for other health and social care professionals to be more aware of their own unconscious bias about alcohol:
*“They might have an unconscious bias and they may judge somebody with an alcohol problem and dismiss them rather than investigate. That is of concern as well. So, it’s about reducing stigma and making people aware of their own unconscious bias, I think about alcohol. It’s just recognising them and its, and I think naturally I think alcohol, you know, addictions are going to be one where people have subtle bias even if they don’t recognise the stigma.”*(Participant #11; Psychiatrist; Drug and Alcohol Services)

Theme 2: No dedicated service but silos with revolving doors.

Throughout the interviews is became very clear that there were no dedicated services for people with ARBD; therefore, there was often little or no signposting for health and social care professionals on where people with ARBD could access treatment or support:
*“…they don’t fit the criteria for any service quite often, you know? Trying to fit them into a pathway when there is no clear pathway for them, it’s very difficult.”*(Participant #11; Psychiatrist; Drug and Alcohol Services)

Many health and social care professionals reported that the type of care and support available to people with ARBD was dictated by where they lived. This postcode lottery meant that the possibility for, and the quality of, diagnosis, treatment and care was largely dependent on the catchment area of the organizations:
*“It depends what area you live, what you have access to. So, I think it’s all about what’s in your area, what you’ve got access to, what you know you’ve got access to, whether they’ve got a bed, you know, it’s just complicated…”*(Participant #22; Clinical Psychologist (Neuropsychology); NHS)

This resulted in some people with ARBD refusing poor-quality care:
*“Some people would rather be on the streets than go to (location of organization).”*(Participant #18; Support Worker; Housing for Homeless)

Key steps in the treatment pathway were reported to be missing, causing revolving door patients where people with ARBD would repeatedly enter and leave a variety of health and social care systems with no continuity of care. This revolving door situation increased the likelihood of people with ARBD dropping out of the system. This lack of continuity of care was understandably linked to people with ARBD relapsing and continuing to consume alcohol at harmful levels:
*“Honestly, a lot of it’s a revolving door. They’ll go, they’ll be evicted from this agency. Go into another one, go into another one and come back round. You get the odd success but few and far between.”*(Participant #14; Senior Support Worker; Health Centre for Homeless)

These reported problems in the health and social care system resulted in people with ARBD being sent to many different organizations primarily without any further support or follow-up and health and social care professionals reported working in silos. The lack of continuity of care was exacerbated by a lack of communication between organizations:
*“…to have an alcohol client looked at toward mental health services is difficult. I find that very difficult because someone without experience with addictions will just look at that person and their diagnosis is, well, they just drink too much.”*(Participant #6; Recovery Coordinator; Addiction Services)

Furthermore, health and social care professionals reported gatekeepers with a lack of knowledge of ARBD were significant barriers to people with ARBD entering the system:
*“So there may be counsellors but they don’t necessarily understand the medical side of things, which can be quite tricky if they’re having to refer people in and we’ve seen, we have seen less alcohol patients coming through to us, than we would have dealt with in the past…having nonclinical services referring patients into our service is always going to make that transition difficult.”*(Participant #11; Psychiatrist; Drug and Alcohol Services)

Theme 3: Underfunded generic care with long waiting lists typically excludes those with alcohol or neurological problems.

Inextricably linked to the finding that there was no or little signposting for either health and social care professionals or for people with ARBD on where to access diagnosis, treatment or support was the fact that there was a lack of care provided specifically for people with ARBD. The little care that was potentially available for people with ARBD had long waiting lists which could exacerbate their problems:
*“So even if you do get a patient referral accepted, quite often patients are waiting months and months before they’re seen. Um, and when people’s mental health needs go untreated for that amount of time, quite often in result of self-medication, which may be in the form of alcohol and drugs.”*(Participant #24; Clinical Psychologist; Acquired Brain Injury Rehabilitation Support Service)

In addition, services maintained strict exclusion criteria, refusing to admit people with complex needs including addiction, alcohol use, psychiatric and neurological problems and organizations typically only dealt with a single aspect of an individual’s problem:
*“…the mental health services have very, very low tolerance for that [alcohol].”*(Participant #25; Clinical Psychologist; Acquired Brain Injury Rehabilitation Support Service)

However, some health professionals reported having found ways around the system:
*“So technically we’re a neuro rehab team, so if someone has a diagnosis of Korsakoff’s..., really, they wouldn’t technically fall within our remit. Although if that person has a diagnosis of Korsakoff’s and then has a fall and has a trauma, even if it’s a mild one we would, we would see them, just as we would see someone who had a fall and has a dementia diagnosis.”*(Participant #23; Clinical Psychologist; Acquired Brain Injury Rehabilitation Support Service)

The main reason cited by health and social care professionals for the lack of care for people with ARBD was a lack of funding:
*“There’s no drug or alcohol rehabilitation units left in the Northeast. They closed down…Funding, austerity, conservative government…It’s a council has lost, one of the biggest percentages in the country in austerity. Of their funding…we’re in the top 5% of cuts that have been made into an area that’s already socio-economically deprived.”*(Participant #3; Social Worker (Substance Use); Council)

This lack of funding resulted in a higher caseload due to a lack of beds in living facilities and specialised care. Long-term care was usually provided in the form of older people’s care homes or religious care organizations which could result in inappropriate long-term care:
*“I think a lot of people who maybe would have a diagnosis of Korsakoff who couldn’t live on their own anymore, could go into nursing homes everywhere and drink.”*(Participant #22; Clinical (Neuropsychology); NHS)

Theme 4: Alcohol-related brain damage remained underdiagnosed and misdiagnosed.

Many health and social care professionals reported relatively low numbers of cases of ARBD being diagnosed within their services although often this was because it was one of the exclusion criteria for their service:
*“It’s low numbers because it’s one of the…one of the exclusion criteria for our service. So, if somebody has got, if they’ve got complex needs, for example, if they’re still currently drinking, there’ll be seen by community treatment teams, either old age or general adult teams and if it’s a suspected Korsakoff’s, they will be seen by neuro-psychiatry colleagues.”*(Participant #20; Consultant Old Age Psychiatrist; Memory Assessment and Management Services)

While some professionals reported seeing “quite a high rate”, they reiterated the lack of dedicated services for people with ARBD:
*“We encounter it in a significant minority of our patients. Probably I would estimate roughly around a third of our patients, probably. So, it’s quite a high rate. It’s not always the presenting factor. So, we usually see someone because of a head injury or brain injury, caused by assault injury, infection, disease…but we also, we also pick up people who have addiction problems as part of that. And a big gap for services at the moment is that they don’t have anything for Korsakoff specifically. So that’s a major issue.”*(Participant #23; Clinical Psychologist; Acquired Brain Injury Rehabilitation Support Service)

Participants described difficulties in diagnosing ARBD, as the symptoms could be attributable to other factors. Participants described many comorbidities that a person with ARBD could present with in addition to the symptoms of ARBD. These complexities related to psychiatric conditions, behavioural conditions, neurological damage, trauma and addiction. Participants explained how symptoms that related to changes in a patient’s behaviour could be difficult to pinpoint for those that had a complex medical history:
*“Quite often a lot of our patients have other comorbidities like other mental health problems like we were talking about precedes Korsakoff’s as well. Whether that’s depression or psychosis or something else on personality disorder that will, when they have Korsakoff’s can come to the forefront a bit sometimes, you know? Especially personality dysfunction sort of Korsakoff...they get coarsening of their frontal lobes and their filter’s gone. They can be more difficult to manage than maybe some other, some other patients.”*(Participant #11; Psychiatrist; Drug and Alcohol Services)

Health and social care professionals reported that one of the reasons for not using a formal diagnosis of ARBD was because this might prevent the person with ARBD from receiving some forms of treatment and or care:
*“I don’t think it would make any difference to what we would do. If anything, it might present a reason for exclusion from this team.”*(Participant #23; Clinical Psychologist; Acquired Brain Injury Rehabilitation Support Service)

Health and social care professionals reported issues that people with ARBD faced when they were misdiagnosed and the importance of a correct diagnosis as beneficial for further treatment:
*“…where we have Korsakoff…because the relatives are concerned that they’ve brought them along, or they’re being misidentified as, I don’t know, delirious or confused or being put into a nursing home or, you know...I think, I think is a hugely problematic area.”*(Participant #24; Clinical Psychologist; Acquired Brain Injury Rehabilitation Support Service)

Those health professionals who regularly diagnosed people with ARBD reported the importance of the umbrella term as it allowed for a clear diagnosis with a complex symptomatology:
*“Alcohol-related brain damage because it encompasses alcohol, nutritional, traumatic um, some of them may have liver disease, may have some encephalopathy, some of them may have coagulopathy as part of the liver disease. So, all of those things can put a strain on cognitive function. Um, so, you know, I guess it is a whole, a whole host of different things feeding into one.”*(Participant #19; Consultant Hepatologist and Alcohol Lead; NHS)

## 5. Discussion

### 5.1. Summary of Findings

Health and social care professionals in our study reported that people with ARBD were stigmatised both by society and professionals which inhibited entry into the health and social care system; therefore, ARBD remained underdiagnosed and misdiagnosed. A lack of diagnosis was sometimes intentional so that people with ARBD could receive treatment as there was no dedicated service for people with ARBD but silos with revolving doors and underfunded generic care with long waiting lists typically excluding those with alcohol or neurological problems.

### 5.2. Comparison to the Literature

The fact that people with ARBD experience significant stigma is not surprising given that the literature has highlighted barriers to treatment and negative attitudes to alcohol use and alcohol-related disorders in general [7], with some healthcare professionals of the opinion that alcohol-related disorders are self-inflicted [7]. In our study, health and social care professionals felt that other professionals held stigmatised views towards people with ARBD, although these were often unconscious biases. Unconscious bias describes associations that reflexively alter perceptions, thereby affecting behaviour, interaction, and decision making [29]. Bias may play an important role in persisting healthcare disparities and unconsciously influence the way information about an individual is processed, leading to unintended disparities that have real consequences in patient care. Fortunately, unconscious bias training has been shown to be effective at both raising awareness and reducing unconscious bias [30].

However, people with ARBD experience dual stigma of both an alcohol use disorder in combination with a cognitive impairment as corroborated by the literature [18,19]. Unfortunately, this combination of health problems resulted in people with ARBD being reportedly excluded from the vast majority of services where people with ARBD might present for diagnosis, treatment and care. Schölin et al. [19] have previously suggested that staff supporting this population who live with the dual stigma of cognitive impairment and alcohol problems require education and training in order to meet patient/client needs. Much of this education should focus on understanding the experiences and perspectives of those living with ARBD in order to inform approaches to care. Developing innovative ways of engaging with complex health and social care needs, promoting a rehabilitative framework and initiating person-centred approaches to care have the potential to facilitate both recovery and improved quality of life in this vulnerable population [19].

The literature highlights other groups of individuals who have been excluded or discouraged from attending health services, in particular people experiencing homelessness [31]. This is extremely concerning considering many of the health and social care professionals who were identified as being involved with people with ARBD in our study were from housing organizations or charities for the homeless. This could present a scenario where individuals are experiencing the triple stigma of alcohol use, cognitive impairment and homelessness. Thomson et al. [7] have called for a sustained campaign against such stigma within the medical profession, which denies access to effective preventive interventions for those most in need. Patients with ARBD should be treated with the same level of care and respect as those who acquire brain damage from other causes such as Alzheimer’s disease or traumatic brain injury [7].

The fact that there are multiple revolving doors along the pathway to receiving diagnosis, treatment and care for people with ARBD was made clear in our study with health and social care professionals. Among people with complex needs and addiction problems the revolving door is a well-known phenomenon [32,33]. Research suggests that there are approximately 60,000 people across England facing multiple and complex needs, with many more at risk of entering this situation. Many in this group are failed by mainstream services as they are excluded for disruptive behaviour or because they do not meet rigid and complicated thresholds for access. Therefore, effective responses to service users with complex needs are needed [34].

It is clear from our work that health and social care professionals experience a lack of funding for drug and alcohol services in the region which serves to exacerbate health inequalities despite the government’s long-term commitment to tackling them [35]. Northeast England experiences some of the worst health inequalities in the country [36] and these health inequalities are only becoming worse with the gap between Northeast England and other regions (particularly the South of England) worsening. Those living in Northeast England are more likely to have a shorter lifespan and to spend a larger proportion of their lives in poor health, as well as being more likely to die prematurely from preventable diseases [36]. In addition, the most disadvantaged in society suffer the most severe consequences from excessive alcohol consumption [7]. It is reassuring that the findings from our study in Northeast England confirm those of Heirene et al., who interviewed 17 Welsh health and social care professionals supporting people with alcohol-related neurocognitive disorders (ARND), who reported the challenging behaviours associated with people with ARND, their significant physical and mental health needs, stigmatisation from others and a general lack of understanding and awareness of ARND [23].

### 5.3. Strengths and Limitations

This is the first study to have examined the pathway for people with ARBD to receipt of diagnosis, treatment and care from the perspective of health and social care professionals in Northeast England. However, there are some weaknesses associated with the research which must be acknowledged. COVID-19 significantly affected data collection putting increased pressure on health and social care professionals and limiting their availability to participate in research. Therefore, despite extensive attempts at recruitment some key stakeholders were not represented in this research including general practitioners, paramedics and accident and emergency staff. Although primary healthcare professionals lacked representation in the final sample data power was felt to have been achieved. In addition, this research also consisted of a relatively small number of participants, although this is the norm when depth of data is required, and people with ARBD were not included in the sample so as not to cause unnecessary stress. The research sample was also limited to Northeast England. However, due to the high levels of alcohol consumption and associated hospital admissions [1], this was a particularly relevant setting for our research. Finally, there was significant variation in the length of interviews with the shortest lasting only 15 min. Shorter interviews occurred where health and social care professionals had less knowledge and experience with people with ARBD, for example in services where a history of alcohol use of any kind was a contraindication. However, shorter interviews were important in revealing this lack of experience of professional’s involved along the journey for people with ARBD and highlighted an opportunity for further signposting, training and development.

## 6. Conclusions

Health and social care professionals need to be challenged on the assumption that ARBD is self-inflicted via training aimed at challenging unconscious bias. Reducing the stigma and stigmatising processes associated with ARBD could counteract health and social care professionals’ reluctance to care for people with ARBD. Ultimately, there is a need for dedicated services for people with ARBD with signposting for health and social care professionals on where people with ARBD can access treatment and support.

## Figures and Tables

**Table 1 ijerph-21-00010-t001:** Example sub-theme from conditional matrix.

Overarching Theme	Sub-Theme	Description	Quotations
Barriers	Stigma	Health and social care professionals report how people with ARBD are severely stigmatised. This stigma is not only exhibited by society but also, by the professionals that are supposed to care for them. Stigma is identified by many health and social care professionals as a barrier for people with ARBD to receipt of appropriate diagnosis, treatment, and care. As a result, despite there being many entry points into the health and social care system for people with ARBD, some would be denied treatment. Some health and social care professionals report that stigma causes people with ARBD to be less inclined to admit to having a problem and therefore less inclined to seek help. In addition, some health and social care professionals describe how people with ARBD could become isolated and abandoned. Those health and social care professionals who regularly work with people with ARBD call for other health and social care professionals to be more aware of their own unconscious bias about alcohol.	Um, because I think it’s about stigma and discrimination. Um, I worked with a client that subsequently didn’t turn out he had Korsakoff’s, but for 10 years he suffered extreme memory issues backwards and forwards to general practitioner was told when you’ve stopped drinking, your memory issues will subside. Um, which left him in an incredibly vulnerable position. I ended up placing him in a residential detoxification and rehabilitation unit, um, where after three months there was no significant improvement with his memory. I asked for some medical investigations. It turned out he had a brain tumour. What’s had been undiagnosed for years. However, because he drank for 37 years, there’s still no clear understanding whether there is some Alcohol Related Brain Damage there as well. Uh, he hasn’t drank for five years now, uh, at all. He still has memory issues. Yeah. #3 Oh yes sir. It may start off more of a hand holding where they will need a bit of extra support to get to appointments, get to groups. Even just address sort of the physical needs getting them to GP’s, etc. As I was saying before about, I genuinely believe this is the most stigmatised group within the social care sector. I’ve worked at a different team a few years ago and worked across all the sectors when I was in mental health and I felt that at the time that was quite stigmatised, quite discriminated against. I had no idea what stigma and discrimination were until I came to work purely drug and alcohol. It doesn’t have any social sympathy attached to it. Which again, I think it creates more problems when you’re talking to them. The hidden population. So we may not get people, it’s very hard to do preventative work with people because they don’t tend to come through our doors. It’s highly problematic. And they’re in the stages of addiction. We look at a neuro-, bio-, psycho-, social-model. So we look at what’s happening with addiction and the brain, which I think is quite a contemporary approach, there haven’t been a lot of research done around. #3

#3 refers to Participant #3; Social Worker (Substance Use); Council.

**Table 2 ijerph-21-00010-t002:** Participant organization type and role.

ID	Group Interview	Organization Type	Role
1	N	Religious Charity	Volunteer
2	N	Council	Social Worker
3	N	Council	Social Worker (Substance Use)
4	N	Council	Assessor
5	N	Public Health/Council	Service User Involvement Officer
6	N	Addiction Service	Recovery Coordinator
7	Y Group 1	Addiction Service	Recovery Coordinator
8	Y Group 1	Addiction Service	Recovery Coordinator
9	N	Addiction Service	Recovery Coordinator
10	N	Addiction Service	Nurse (Substance Use)
11	Y Group 2	Drug and Alcohol Service	Psychiatrist
12	Y Group 2	Drug and Alcohol Service	Medical Student
13	Y Group 2	Drug and Alcohol Service	Medical Student
14	Y Group 3	Health Centre for Homeless	Senior Support Worker
15	Y Group 3	Health Centre for Homeless	Senior Support Worker
16	Y Group 3	Direct Access Hostel	Senior Support Worker
17	Y Group 3	Housing for Homeless	Engagement Worker
18	Y Group 3	Housing for Homeless	Support Worker
19	N	National Health Service	Consultant Hepatologist/Alcohol Lead
20	N	Memory Assessment and Management Service	Consultant Old Age Psychiatrist
21	N	Centre for Neuro Rehabilitation and Neuropsychiatry	Consultant Neuropsychiatrist
22	N	National Health Service	Clinical Psychologist (Neuropsychology)
23	Y Group 4	Acquired Brain Injury Rehabilitation Support Services	Clinical Psychologist
24	Y Group 4	Acquired Brain Injury Rehabilitation Support Services	Clinical Psychologist
25	Y Group 4	Acquired Brain Injury Rehabilitation Support Services	Clinical Psychologist

## Data Availability

Participants provided written informed consent to participate in this study. In line with the terms of consent to which participants agreed, the data are not publicly available. There are ethical restrictions on sharing the de-identified dataset. The data contain potentially identifying and sensitive participant information and we do not have participant consent to share this dataset. Data requests may be sent to Northumbria University Research Ethics Committee (ref number 9271).

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
