# Peer review of "“I Genuinely Believe This Is the Most Stigmatised Group within the Social Care Sector”—Health and Social Care Professionals’ Experiences of Working with People with Alcohol-Related Brain Damage: A Qualitative Interview Study"

_ijerph, 2023, doi:10.3390/ijerph21010010_

Round 1

Reviewer 1 Report

Comments and Suggestions for Authors

Dear Authors & Editorial Team,

Many thanks for the invite to review this interesting manuscript. Please see my comments on each of the sections below.

Abstract

“Professionals need to be challenged on the assumption that Alcohol Related Brain Damage is self-inflicted via training aimed at challenging unconscious bias.” – This statement seems confrontational and the final part rather definitive, as if unconscious bias training is an unequivocally effective strategy for reducing stigma (it’s not and therefore this seems like a big leap to make from the data in this study, which say nothing of this training). The authors don’t provide a single study anywhere in the manuscript that supports the value of unconscious bias training (only fleetingly mentioning it in the final concluding paragraph), and yet it has a prominent place here in the abstract (the only thing we know many people will read).

Introduction

“The number of people in the North-East of England with ARBD is hard to determine, partly due to outdated epidemiological figures, but also as a result of a failure to diagnose the condition [6].” – Have any prevalence studies been conducted in this region? If so, they should be cited, but if not this statement should be altered. There was a prevalence study of ARBD among the homeless in Scotland, and our study of the prevalence in South Wales, both of which seem appropriate to mention here given the close location and recency of the latter. [I reference a study I have published as I believe it is genuinely relevant to the publication. If the authors do not wish to cite this paper I will have no concerns and this would not in any way affect my review or recommendations]

  • Gilchrist G, Morrison DS. Prevalence of alcohol related brain damage among homeless hostel dwellers in Glasgow. Eur J Public Health. 2005 Dec;15(6):587-8. doi: 10.1093/eurpub/cki036. Epub 2005 Sep 14. PMID: 16162595.
  • Heirene R, Roderique-Davies G, Angelakis I, John B. Alcohol-related neurocognitive disorders: a naturalistic study of nosology and estimation of prevalence in South Wales, United Kingdom. J Stud Alcohol Drugs. 2020;81:584–94.

It is quite surprising to me that the authors did not discuss our recent study that involved interviewing 17 diverse health professionals involved in supporting those with alcohol related brain damage working in Wales (titled: Professional Perspectives on Supporting Those with Alcohol-Related Neurocognitive Disorders: Challenges & Effective Treatment). Nearly every issue and challenge discussed in the introduction is addressed in our paper in quite some detail, and the study used a very similar methodology to that outlined here. [again, while I reference a study I have published as I believe it is genuinely relevant to this publication, if the authors do not wish to mention or cite this paper I will have no concerns and this would not in any way affect my review or recommendations]

There is also another 2006 report that involved qualitative interviews with healthcare staff supporting those with ARBD and a small qualitative study with individuals with Korsakoff’s syndrome (both cited below). Further, the authors state “There is, however, a paucity of literature addressing either experiences of people living with ARBD, or of those providing services and support to that population. In their review, Schölin et al. concluded that understanding these life experiences should be a research priority in order that future service provision, support, and interaction can effectively meet the needs of individuals and their families.”

Combined, the studies mentioned above and cited below challenge the assertions made by the authors here; there is still clear need for qualitative research on this subject like that reported here, but it doesn’t need to be justified by saying there is no existing research like this.

  • Heirene, R. M., John, B., O’Hanrahan, M., Angelakis, I. & Roderique-Davies, G. (2021). Professional Perspectives on Supporting Those with Alcohol-Related Neurocognitive Disorders: Challenges & Effective Treatment. Alcoholism Treatment Quarterly, 39 (3), 1–27. https://doi.org/10.1080/07347324.2021.1898294
  • McCabe, L. (2006). Working with people with alcohol-related brain damage.  Stirling, UK: http://www.dldocs.stir.ac.uk/documents/workingwitharbd.pdf
  • van den Hooff, S. L., & Goossensen, A. (2015b). Ethical considerations on the value of patient knowledge in long-term care: A qualitative study of patients suffering from Korsakoff ’s syndrome. Nursing Ethics, 22(3), 377-388. http://dx.doi.org/10.1177/0969733014534876

Methods

This section is mostly very clear, well written, and contains all relevant details. However, I would like to see the authors expand upon the steps taken during the analysis. At present, the steps are very briefly described, with little relation to the actual data collected in this study. For example, what did the authors actually do in the process of open coding? What types of concepts were identified and what is meant by their properties and dimensions? Can the entire conditional matrix or a snippet of it be shared (see example on page 6 here: https://robheirene.netlify.app/publication/2021-perspectives-on-arbd-treatment-heirene/Heirene_et_al_(2021)_Perspectives_on_ARBD_treatment.pdf)?

Results

Table 1: the authors may want to consider whether reporting individual level data, including gender, age, organisation, role, and years of experience may unintentionally reveal the identities of any interviewees?? Particularly given that direct quotes are linked to interviewees that can be traced back to this table (the type of employer is also mentioned next to quotes; e.g., NHS, council, which adds further concerns regarding anonymity). I know this was a concern in our qualitative study as, for example, there are very few clinical psychologist involved in addiction services in Wales, and most (of the very few) people interested enough in ARBD to be involved in the study were well known in the region.

The qualitative findings are nicely summarised, with illustrative quotes provided. The themes identified are very similar to those identified in our qualitative study. I don’t see this as an issue, I think it’s good to have several studies that converge on similar themes, but it does further highlights the relevance of our paper to this one.

  • Heirene, R. M., John, B., O’Hanrahan, M., Angelakis, I. & Roderique-Davies, G. (2021). Professional Perspectives on Supporting Those with Alcohol-Related Neurocognitive Disorders: Challenges & Effective Treatment. Alcoholism Treatment Quarterly, 39 (3), 1–27. https://doi.org/10.1080/07347324.2021.1898294

Discussion

I have no concerns with this section, other than the lack of the relevant studies highlighted in the introduction and the issues surrounding unconscious bias training mentioned in my comments on the abstract.

Overall comments

I hope the authors find my comments useful in revising their manuscript.

Kind regards.

Reviewer 2 Report

Comments and Suggestions for Authors

The article’s theme (ARBD) is unfortunately a social and health problem for the individuals and their social contexts (family, work, friends), and for the social and health sectors that are responsible for the delivery of care. Based on a local (empirical) reality (“North-East of England large amounts of alcohol are consumed, reflected in the number of alcohol-related deaths in the region”, lines 13-14), they aim to understand how the professionals’ stigma can affect the care delivery. However, the article has no scientific discussion about the complexity of the subject of study. The qualitative study is expected to do a critical analysis of the subject: why this population is stigmatized by professionals? Is it because of the lack of training, therapeutics, knowledge, and funding? Is there any association with “cultural monitories” or “classism”? Indeed, the authors initiated with the idea that “some clinicians holding the view that alcohol-related illnesses are self-inflicted and hence are not worthy of medical care” (lines 46-47); and they concluded the article with the same idea: “Health and social care professionals need to be challenged on the assumption that ARBD is self-inflicted via training aimed at challenging unconscious bias. Reducing the stigma and stigmatising processes associated with ARBD could counteract health and social care professionals’ reluctance to care for people with ARBD” (lines 250-253). After an analysis and discussion of data, keeping the same idea in the end is not a correct approach scientifically and ethically. Personally, I remain with the idea that social and health professionals are not competent to carry out their work, and nothing has been said about the environmental conditions of work, the complex trajectories of those individuals, and the clinical and behavioural conditions of the disease.

Suggestions:

Improve the introduction with information about the “stigma” (it’s missing a conceptualization of this behaviour), the disease (ARBD), clinical and behavioural, the individual trajectories of the people with ARBD, and about the Health&Care system (organizations, pathways, funding, limitations).

 I suggest including also numbers about this population in the UK and Worldly. Using the “North-East of England large amounts of alcohol are consumed, reflected in the number of alcohol-related deaths in the region” is like a journalist approach, like a “the newspaper headline”. The introduction (as a background and a literature review) must support a set of analytical frameworks (categories and subcategories) that there isn’t in the article. These categories are used in the discussion but without theoretical support or justification. Why do authors select those categories?

Improve the materials and methods. What were the questions addressed in the interview? What was the theoretical support of those questions? How do the authors explain the big difference in the length of in-depth interviews (between 15 min. and 90 min)? It’s necessary to detail how the authors prepared and did the interview to understand if the interviews were “in-depth” and “semi-structured”. It’s missing the method of analysis (content analysis, descriptive, adopting software or not?)

Results/Discussion: There is a lot of important information provided by the authors to analyse the complexity of the subject (ARBD). I didn't find any empirical evidence that professionals stigmatised the subject (ARBD); I found a set of external conditions that could be classified as stigmatising (e.g., unawareness, care models/disintegration between health specializations, social ties and environment).

Missing a conclusion that provides next steps, study limitations, consequences for the health and social sectors, required policies (social, health, education, social services).

Round 2

Reviewer 1 Report

Comments and Suggestions for Authors

This is a much improved manuscript and I’m looking forward to seeing this published. I have just a one concern:

  • Statement in the abstract: “Unconscious bias training shows promise for challenging professional’s assumptions that Alcohol Related Brain Damage is self-inflicted” – Again, whilst the authors of toned this down, this statement in the abstract implies that there is evidence that unconscious bias training has been used to reduce stigma towards those with ARBD successfully, which it hasn’t (to my knowledge). It also implies that the authors may have found evidence to support this contention, or discussed it in quite some depth in the paper, which is not the case. The authors have added to their discussion a line that cites one 2018 (non-peer reviewed) report which is very hesitant about the benefits of unconscious bias training and does not discuss its benefits for reducing stigma related to ARBD or alcohol use in the absence of cognitive impairment. Here is the executive summary from the report:

UBT is effective for awareness raising by using an IAT (followed by a debrief) or more advanced training designs such as interactive workshops. UBT can be effective for reducing implicit bias, but it is unlikely to eliminate it. UBT interventions are not generally designed to reduce explicit bias and those that do aim to do so have yielded mixed results. Using the IAT and educating participants on unconscious bias theory is likely to increase awareness of and reduce implicit bias. The evidence for UBT’s ability effectively to change behaviour is limited. Most of the evidence reviewed did not use valid measures of behaviour change. There is potential for back-firing effects when UBT participants are exposed to information that suggests stereotypes and biases are unchangeable. Evidence from the perspective of the subjects of bias, such as those with protected characteristics, is limited. This evidence could provide additional information on potential back-firing effects.

This is a good study, the authors don’t need to make conclusions that go beyond the data. This statement is simply misleading and unnecessary. The only way something like this could be included in the abstract for this paper would be to say something to the effect of: “One potential way of reducing the stigma associated with ARBD may be unconscious bias training, although this has not been evaluated for this purpose”… which just seems odd and speculative. If the sentence were removed, the abstract would read well (the final sentence included).

  • Minor: on page 3, line 110, there is a spelling mistake: “3tigmatization from others”

Author Response

Thank you for these comments we have removed the sentence referring to unconscious bias training from the abstract completely and have corrected the spelling mistake.

Reviewer 2 Report

Comments and Suggestions for Authors

Dear authors,

I accept your review. But, I would like to share two comments.

- The subject of study could be clarified: Is the stigma? Is the perception on another group's stigma? Is the influence of the professionals' stigma in the patients' pathways in health&care systems?

- The methodology should be identified: Is a case study? Is an ethnography? Is descriptive or comparative?

Good luck. I will like to follow your work in the future. The topic is relevant.

Author Response

Thank you for these further comments the paper relates to health and social care
professionals experiences of dealing with people with alcohol related brain damage, as stated in the title. These professionals report (in the results section) how, in their view, people with alcohol related brain damage are stigmatised both by society in general and by other health and social care professionals and how this effects entry into the health and social care system.

The methodology is also presented within the title: A qualitative interview study.